# Obesity and its associated risk factors among school-aged children in Sharjah, UAE

**Abduelmula R. Abduelkarem**[1]*, **Suleiman I. Sharif**[1], **Farah G. Bankessli**[1], **Sherin A. Kamal**[1], **Nahed M. Kulhasan**[1], **Amar M. Hamrouni**[2]

**1** Pharmacy Practice and Pharmacoetherapeutics Department, College of Pharmacy, University of Sharjah, Sharjah, United Arab Emirates, **2** College of Pharmacy, Al Ain University, Al Ain, United Arab Emirates

* aabdelkarim@Sharjah.ac.ae

**Data Availability Statement:** All relevant data are within the manuscript and its Supporting Information files.

## Abstract

### Background

The most prevalent nutritional disorders worldwide are childhood overweight or obesity. Various factors clearly contribute to the childhood obesity epidemic. The aim of this study is to investigate the prevalence of childhood obesity in children of primary schools, and determine the influence of eating behavior and lifestyle in such a condition.

### Methods

The study based on a cross sectional survey including school children aged 6–11 years. Pupils were from different schools in Sharjah, UAE. Outcome measures used in this study covered health characteristics; child habits and lifestyle; disease status and medication.

### Results

The number of pre-validated surveys distributed was 932 and those returned counted to 678, giving a response rate of 72.8%. More than half (379; 55.9%) of the participants were females and 191 (28.2%) of the children were obese or overweight. Almost one quarter (162; 23.9%) of the children was physically inactive. Additionally, candy and fast food consumption was significantly high (370; 54.6%) and (324; 47.8%) respectively. Participant's food, age and time spent on TV were significantly associated with body mass index (BMI).

### Conclusion

Prevalence of overweight and obesity in the Emirate of Sharjah is high in both genders and across all ages of the study population. Contributing factors may include; sedentary lifestyle, consumption of unhealthy food and family history. There is a need for an immediate attention and measures to reduce the prevalence of obesity and associated diseases.

**Funding:** The author(s) received no specific funding for this work.

**Competing interests:** The authors have declared that no competing interests exist.

## Introduction

Obesity refers to the excessive accumulation of fat in the body, which leads to co-morbidities that negatively affect the obese person's health [1]. The Centers for Disease Control and Prevention (CDC) defined overweight as body mass index (BMI) over 85[th] and below 95[th] percentile and obesity as a BMI over 95[th] percentile [2]. The most prevalent nutritional disorders in the United States (US) are childhood overweight and obesity. In recent years, the occurrence of childhood obesity has tripled in the US as stated by The Obesity Society [3]. In the United Arab Emirates (UAE), the prevalence of overweight and obesity varied across age, where below 9 years of age both sexes were below the international standards with an increase overweight and obesity occurring among both genders of 9 to 18 years [4]. A more recent study demonstrated that in UAE, obesity starts in toddlers and progresses linearly with age and pointed out an alarming high prevalence of extreme obesity, especially among boys [5].

The estimated worldwide prevalence of childhood overweight and obesity has increased from 4.2% in 1990 to 6.7% in 2010, and it is expected to rise more in the following years to reach up to 9.1% in 2020 [6].

A combination of various factors clearly contributes to the childhood obesity epidemic. Hence, knowledge of these causative factors will help in the prevention of obesity. Socioeconomic and psychological factors may also contribute to childhood obesity [7]. The consumption of high-calories food with no or limited physical activity is the main contributor to childhood obesity [8]. Additionally, psychological factors such as familial stress, anxiety, and social isolation may contribute to childhood obesity. Children tend to increase their consumption of food to deal with their emotions and problems [8]. In addition, some people have a genetic predisposition for being overweight or obese, but most do not become overweight or obese unless there is an imbalance between calories consumed in diet and calories burned [9]. Some studies found that children of lower socioeconomic status have lack of safe places for physical activity and limited access to healthy food thus they tend to consume food that does not spoil quickly such as frozen meals, in addition to high intake of fast and fried food [10]. Other studies mentioned that with the economic improvement, a greater demand for fat-rich diet results in an increase in energy consumption predisposing children to obesity and other diseases [11]. Moreover, obese children are at higher risk of developing cardiovascular diseases, type 2 diabetes and certain types of cancer such as endometrial, colon, and postmenopausal breast cancer [12]. Prevention is the key element for controlling childhood obesity. Strategies that can be followed including; the primary prevention of overweight or obesity itself and second is the prevention of weight regain after weight loss, plus prevention of weight gain in individuals unable to lose weight [7]. Several studies have explored childhood obesity [4, 12], and without doubt, the problem is an issue of public concern. There is also unanimous view emphasizing that the paramount increase in childhood as well as adulthood obesity requires continuous surveillance [13, 14]. Providing recent estimates of childhood obesity is essential to keep the focus on this matter as it has many consequences on the children health [7]. Based on this background, the aim of this study was to investigate the prevalence of childhood obesity in various schools in Sharjah, UAE, and to determine the influence of eating behavior and lifestyle of children on obesity.

## Methods

### Study design

The present study was conducted in Sharjah, UAE, based on a cross sectional survey over a period of four months (August to November 2017).

## Ethical approval

This study received an ethical approval from the University of Sharjah Ethical Committee (REC-17-09-10-01-S). We informed parents, teachers, and the supervisors about the purpose and the nature of the study. The survey was coded, and the names were kept anonymous.

## Questionnaire development

The development of questionnaire was based on the information needed for the study and it was written in both Arabic and English languages. The questionnaire was pre-piloted for face validity by distributing it to 12 parents of school children of the targeted age and their comments and recommendation were taken into consideration in the final version of the survey but their responses were included in results. The questionnaire included 20 items covered in two sections. The first section covered questions to collect information on socio-demographic and health related characteristics which included nine items (gender, age, ethnicity, weight, height, BMI percentile, family history of obesity, disease status and medication intake). The second section consisted of eleven items to collect information about child eating habits and lifestyle, and questions in this section addressed the child's eating patterns. These include the source of food consumed at school; frequency of consumption of fast food and candy, number of healthy meals, type and nature of food consumed as well as number of total meals per day. In addition, the section included questions on lifestyle (time interval between bedtime and the last meal, time spent using electronic devices, and physical activity). In this context we defined physical activity as any indoor or outdoor sport activity not including the use of smart devices or watching TV. On the other hand physical inactivity is defined as sedentary pleasurable activities such as watching cartoons on TV or using smart devices for gaming.

The response options for participants to select from were variable. In questions on time intervals, we used a range of 30 minutes to more than two hours, for frequency (always, often, sometimes, rarely), and "Yes" and "No" answers. For other questions, the options were general for example, when the parents were asked about the ways of preparing foods for their children, the options were "boiled"," fried", "grilled" or "mixed", and for the nature of food consumed, the options were "carbohydrates", "fat", "protein" or "mixed". Additionally, we asked parents about the source of the food consumed by their children at school, and the options were "Bring his/her breakfast from home", "Buy a meal from cafeteria", "Just eat snacks", "Skip breakfast". The total number of surveys distributed was 932.

## Study population

From a list provided by the Ministry of Education, we selected schools in the Emirate of Sharjah based on their accessibility and availability and applied the non- probability sampling method. The study population included both male and female pupils aged 6–11 years. The total representative sample was 678 pupils (299 boys and girls 379) in eight schools in Sharjah; UAE using the online sample size calculator was 248 with a confidence interval of 95% and 5% margin of error for a population of 65473 from 8 schools [15–17]. Within each selected school, pupils from Grade I to V were offered the opportunity to voluntarily and anonymously participate.

We based the selection of the pupils on the resubmission of questionnaire. The exclusion criteria included children with special needs of age below 6 and above 11 years, and parents who refused to participate in the study. Similar to the protocol employed in earlier studies [4], we trained members of the research team for three weeks on obesity, anthropometric measurements and how to conduct structured interviews of the participants.

## Anthropometric measurements

Trained researchers measured student's height and weight in bare feet while wearing light-weight school uniform in the early morning before they start their daily classes. Weight (in kilograms) and height (in centimeters) were measured using Seca 799 scale. The BMI was calculated using Medscape online calculator. We defined obesity according to the CDC guidelines as a BMI $\geq$ 95th percentile and overweight as a BMI $\geq 85^{th}$—$\leq 95^{th}$ percentile. The students were instructed to stand straight with their heads, backs and buttocks vertically aligned to the height gage, and then their heights were taken and rounded to the nearest 0.5 cm. Simultaneously, the students' weights were recorded from the digital screen and rounded to the nearest 0.5 kg. The three most important anthropometric measurements of clinical importance to be taken in paediatric practice are height or supine length, weight and head circumference. These measurements describe different body components and their changes have different biological significance. For other populations other than paediatrics the commonly obtained anthropometric measurements include height, weight, knee height, elbow breadth, triceps skinfold, subscapular skinfold, arm circumference, abdominal circumference and calf circumference. Obtained measurements are compared to standardized percentiles. Anthropometric measurements can be combined in the evaluation of nutritional status. For example, weight for height has been demonstrated as a strong predictor for 12-month mortality in haemodialysis patients. Mortality rate appears to decrease as patient's weight for height increase. BMI is also an important predictor. The use of correct body weight is essential for patient assessment and for determination of dietary needs. A variety of definitions of body weight have been used for nutritional assessment such as usual body weight, standard body weight, and ideal body weight [18]. Anthropometry involves the external measurement of morphologic traits of human beings. High quality anthropometric measurements are fundamental to clinical and epidemiological research. The measurements for each method have inherent variations, either due to biologic variation or due to error in measurement. Errors in measurement cannot be avoided completely but they can be minimized to a large extent. We define methods to estimate measurement error in anthropometry, offer guidelines for acceptable error, and suggest ways to minimize measurement error; thereby improving anthropometry quality in health assessments. We propose that special attention be paid to the following six key parameters for quality assurance of anthropometric measurements: (i) Identification of certified lead anthropometrist and trainer, (ii) manual of standard operating procedures, (iii) choice of robust equipment, (iv) equipment calibration, (v) standardization training and certification, and (vi) measurements resampling [19]. To abide by the above-mentioned parameters of quality assurance of anthropometric measurements, the researchers in this study were trained by certified professionals to take the measurements, and also measurement resampling was done. In addition to that the equipment used were standardized and calibrated before use. This is to improve anthropometric measurements quality and eliminate the error as much as possible. Finally, BMI measurements were calculated accordingly.

## Statistical analysis

We encoded the participants' responses and analyzed the data using Statistical Package for Social Sciences (IBM SPSS statistics for windows, version 20.0, IBM Corp., Armonk, NY, USA). We adopted descriptive analysis to calculate the response proportion of each group of respondents for each item in the questionnaire. We also used the Chi-square test to ascertain the association between the dependent variables and other independent selected variables considering the level of p< 0.05 as the cut-off value of significance.

## Results

The total of surveys distributed was 932 and 678 surveys were completed and returned back giving a response rate of 72.8%. Throughout the process of distribution and collection of surveys, 46 samples were excluded including 4 participants who refused to participate, 9 were above 11 years, and 22 returned back unfilled surveys. We did not receive back additional 212 surveys, as the parents of the pupils were not interested to take a part in the study. Among the study pupils 299 (44.1%) were boys and 379 (55.9%) were girls. The average age of the pupils was 8.2±1.7 years, with an average height of 131±11.1 centimeters and weight of 30.5±10.8 kilograms. Less than one quarter (134; 19.8%) of the participants reported to have a family history of obesity. A total of 487 (59.4%) children showed normal range of BMI whereas 95 (14%) and 96 (14.2%) had an obese BMI and overweight respectively. Table 1 summarizes the socio-demographic and clinical characteristic of participants.

### Eating habits of participants

In the present study, 456 (67.3%) of the pupils included in the study ate their in-home prepared food and 21.1% (143) reported that they buy their meals from the school cafeteria. An interval of 1 to 2 hours elapsing between the last meal consumed and bedtime was reported by 241 (35.5%) of participants. Slightly more than half (372; 54.9%) of the respondents reported their children to consume food of mixed nature (carbohydrates, fats and proteins). The majority (590; 87%) reported to consume food that was prepared by boiling, frying or grilling

**Table 1. Socio-demographic and clinical characteristics of participants.**

| Characteristic | Frequency (%) |
|---|---|
| **Child Gender**: | |
| Female | 379 (55.9) |
| Male | 299 (44.1) |
| Total | 678 (100.0) |
| **Family history of obesity**: | |
| Yes | 134 (19.8) |
| No | 544 (80.2) |
| Total | 678 (100) |
| **Overweight and obesity in pupils with family history** | |
| Overweight | 32 (4.8) |
| Obese | 34 (5) |
| Total | 66 (9.8) |
| **BMI Percentile**: | |
| Underweight | 84 (12.4) |
| Normal weight | 403 (59.4) |
| Overweight | 96 (14.2) |
| Obese | 95 (14.0) |
| Total | 684 (100.0) |
| **Source of food services**: | |
| Bring his/her lunch from home | 456 (67.3) |
| Buy a meal from school cafeteria | 67 (9.9) |
| Just eat snacks. | 143 (143.0) |
| Skip lunch | 12 (1.8) |
| Total | 678 (100.0) |

methods. Consumption of two healthy meals per day was reported by 292 (43.1%) of the participants. Table 2 summarizes the eating habits of children included in the study.

## Participants' leisure time

When the parents were asked about the time their children spend watching TV and using electronic devices, more than one quarter of those watching TV (237, 35%) and using electronic devices (203, 29.9%) reported spending 1–2 hours daily on such a leisure. Furthermore, almost one quarter (162; 23.9%) of the children reported low levels of physical activity. Table 3 summarizes the patterns of children's physical activity and leisure time.

## Disease status and medications use by participants

Only 82 (12.1%) of the children were reported to have a disease and use medications. Iron deficiency anemia, allergy and asthma problems were among the most common diseases reported during the study period. Anti-allergic, anti-asthmatic, and iron supplements were among the most commonly medications used by the children.

## Relationship between BMI percentile, pupil's habits and lifestyle

Using chi-square test, we observed a number of statistically significant associations between BMI percentile, pupil's habits and lifestyle. The number of daily healthy food consumption and the predominant food in the child's diet were both significantly ($p < 0.002$ and $p < 0.001$ respectively) associated with BMI percentile of the participants. Furthermore, there was a significant association ($p < 0.04$) between time spent on television and BMI percentile of pupils included in the study (Tables 4 and 5).

## Discussion

We carried out the present study to determine whether childhood obesity is predominant in the Emirate of Sharjah, UAE and to establish the association between obesity and the child's lifestyle and eating patterns. The global prevalence of overweight and obesity in 2010 was 6.7% [1]. Results of the present study revealed that the prevalence of overweight and obese children in UAE is 28.2%, which is around four times higher than the global prevalence described above for 2010. Our findings suggest that the increase in the prevalence of obesity in Sharjah-school pupils is in total agreement with the results of recent studies carried out in UAE on different population samples [4]. Over the last 5 decades, the economic enhancement in all countries of the Eastern Mediterranean Region (EMR) was associated with a greater demand on diet that is rich in fat. Such a tendency by increasing the intake of energy predisposes to obesity and diseases [14].

Similar to the results of an earlier study in UAE (17), we also identified a high prevalence (12.4%) among school children. About 27.5% of the underweight children were between 6–7 years old. These findings suggest that under nutrition is common among children in the UAE despite the high-income status of the country. We noticed that the lowest consumption of healthy diet was in children who were underweight, which might be an underlying factor. Poor knowledge of healthy dietary patterns may be a reason for under nutrition. However, the exact determinants for under nutrition among children in the UAE warrants further dedicated investigation.

We also observed that 19.8% of the participants had a family history of obesity, and 9.8% of them were overweight and obese. This is consistent with the findings of a similar study conducted in the Emirate of Abu Dhabi [20]. These results taken together, strongly supports the link that is widely documented in literature between family history of obesity and the development of obesity in children. A family history of obesity may promote childhood obesity

**Table 2. Eating habits of participants.**

| Eating Habits | Frequency (%) | |
|---|---|---|
| | N = 678 | |
| **Number of Meals During a Day** | 1 meal | 20 (2.9) |
| | 2–5 meals | 646 (95.3) |
| | > 5 meals | 12 (1.8) |
| | **Total** | 678 (100.0) |
| **Time Interval Between Child Last Meal and Bed Time** | 0-30minutes | 95 (14.0) |
| | 30 minutes—1 hour | 204 (30.1) |
| | 1 hour—2 hour | 241 (35.5) |
| | > 2 hours | 138 (20.4) |
| | **Total** | 678 (100.0) |
| **Type of Food That Child Eats** | Boiled | 18 (2.7) |
| | Fried | 49 (7.2) |
| | Grilled | 21 (3.1) |
| | Mixed | 590 (87.0) |
| | **Total** | 678 (100.0) |
| **Predominant food in child's diet** | Carbohydrate (Rice, Bread, Milk, Yogurt, Corn, Potatoes, Fruits) | 89 (13.1) |
| | Fats (Cheese, Dark Chocolate, Whole Eggs, Olive oil, Nuts) | 11 (1.6) |
| | Proteins (Meat, Chicken, Fish, Eggs) | 206 (30.4) |
| | Mixed | 372 (54.9) |
| | **Total** | 678 (100.0) |
| **Eating Fast Food** | Always | 42 (6.2) |
| | Often | 324 (47.8) |
| | Sometimes | 207 (30.5) |
| | Rare | 105 (15.5) |
| | **Total** | 678 (100.0) |
| **Eating Candy** | Always | 370 (54.6) |
| | Often | 234 (34.5) |
| | Sometimes | 55 (8.1) |
| | Rare | 19 (2.8) |
| | **Total** | 678 (100.0) |
| **Number of Daily Healthy Food (Fruits and Vegetables)** | 0 | 12 (1.8) |
| | 1 | 197 (29.1) |
| | 2 | 292 (43.1) |
| | 3 | 136 (20.1) |
| | > 4 | 41 (6.0) |
| | **Total** | 678 (100.0) |

**Table 3. Patterns of physical activity, daily leisure time and time spent using electronic devises.**

| Criteria | | Frequency (%) |
|---|---|---|
| **Physical Activity (classified per week)** | Always (7 days) | 169 (24.9) |
| | Often (3–5 days) | 226 (33.3) |
| | Sometimes (1–2 days) | 121 (17.8) |
| | Rare (0–1 day) | 162 (23.9) |
| | **Total** | 678 (100.0) |
| **Leisure Time (per day):** | | |
| **Time Spent Watching TV** | 30 minutes | 100 (14.7) |
| | 30–60 minutes | 160 (23.6) |
| | 60–120 minutes | 237 (35.0) |
| | > 120 minutes | 181 (26.7) |
| | Total | 678 (100.0) |
| **Time Spent on Phone, Tablet, and Computer** | None | 21 (3.1) |
| | 30 minutes | 136 (20.1) |
| | 30–60 minutes | 149 (22.0) |
| | 60–120 minutes | 203 (29.9) |
| | > 120 minutes | 169 (24.9) |
| | **Total** | 678 (100.0) |

through environmental and biological mechanisms. With regard to the environmental risk, mother's personal eating habits are likely to influence children's dietary patterns. Biologically, children whose both parents are overweight are predisposed to have a high BMI [21]. However, although genetic factors may contribute to the development of obesity, some researchers suggested that genetic factors only account for up to 5% of childhood obesity cases [22]. A significant association between child's age and BMI percentile (p-value = 0.01) was found. The prevalence of obesity and overweight was the highest (14.2%) in children aged 11 years and lowest (3%) at the age of 7 years. This is consistent with the results of a large cross-sectional study among UAE school children in both the Emirates of Abu Dhabi and Ras Al Khaimah [4, 5]. When taken together, these results reveal a trend of a progressive increase in the

**Table 4. Association of food habits with BMI percentile.**

| | BMI Percentile, Chi square test (p <0.002) | | | | |
|---|---|---|---|---|---|
| **Number of daily healthy food** | **Underweight** | **Normal weight** | **Overweight** | **Obese** | **Total** |
| **0** | 3 | 6 | 2 | 1 | 12 |
| **1** | 30 | 109 | 27 | 31 | 197 |
| **2** | 34 | 178 | 38 | 42 | 292 |
| **3** | 13 | 91 | 13 | 19 | 136 |
| **4 or more** | 4 | 19 | 16 | 2 | 41 |
| **Total** | 84 | 403 | 96 | 95 | 678 |
| | BMI Percentile, Chi square test (p <0.001) | | | | |
| **Predominant food in child's diet** | **Underweight** | **Normal weight** | **Overweight** | **Obese** | **Total** |
| **Carbohydrates** | 14 | 51 | 12 | 12 | 89 |
| **Fats** | 1 | 3 | 1 | 6 | 11 |
| **Proteins** | 19 | 136 | 27 | 24 | 206 |
| **Mixed** | 50 | 213 | 56 | 53 | 372 |
| **Total** | 84 | 403 | | | |

**Table 5. Association of time spent on television with BMI Percentile.**

| Time spent on television/day | BMI Percentile, Chi square test (p < 0.040) | | | | |
|---|---|---|---|---|---|
| | Underweight | Normal weight | Overweight | Obese | Total |
| 30 minutes | 10 | 65 | 16 | 9 | 100 |
| 30 minutes—1 hour | 16 | 113 | 14 | 17 | 160 |
| 1–2 hours | 34 | 125 | 38 | 40 | 237 |
| More than 2 hours | 24 | 100 | 28 | 29 | 181 |
| Total | 84 | 403 | 96 | 95 | 678 |

prevalence of overweight and obesity with age across the population of schoolchildren in the UAE that reflects global trends [23, 24]. The onset of puberty that occurs in children in this age group may contribute to the increase in the prevalence of obesity. Puberty is associated with the accumulation of adipose tissue and decreased physical activity; particularly among children who experience early puberty. These findings indicate that early interventions targeting children aged 6–7 years may be more likely to achieve better outcomes in minimizing or preventing the development of obesity in children.

Results of the present study also revealed that the time spent watching TV and using smart devices was relatively high. About 35% of the total participants spend 1–2 hour per day watching TV and of those 11.5% were obese and overweight. This is consistent with results of the study carried out at the University of Texas Health Science Center and Baylor in Houston, which reported that 67% of their participants watched TV with an average of 1.84 hours per day [23]. Other studies suggested that watching TV has several effects that may lead to obesity including decreased metabolic rate, and increased snacking while watching TV in addition to the influence of food advertisements. Moreover, it has been documented that increasing the time spent on watching TV will increase the prevalence of obesity [24], a finding that parallels results of the present study but contrasts with those of another study that failed to show a link between BMI and watching TV [25].

Additionally, 29.9% of children spent 1–2 hours of their time on smart phones, tablets, and computers but we observed no significant association between the BMI percentile and time spent on these electronic devices. However, there is a correlation between the child's gender and time spent on smart phones and computers where 61.9% of the total male participants spent more than 1 hour on electronic devices as compared to 49.3% of the total female participants. Within the age pool of our study, the fact that boys were more interested in such instruments than girls may explain such results. There was no relationship between physical activity and BMI percentile (p value = 0.443), which is consistent with the findings of other studies that failed to demonstrate a significant effect of physical activity on BMI [24]. Although there is no difference in the obesity predominance between the two genders, we observed a significant association between the physical activity and child's gender in which the number of girls (15.5%) of the total population that rarely exercise is twice that of boys (8.4%). Traditional, cultural, and religious beliefs, as well as the inadequate sport facilities for females in schools are contributing factors for the reduced physical activity and sedentary lifestyle in females. The strict supervision by the parents over their children may also be a reason. The reason behind this strict supervision is that parents prefer to keep their children under their scrutiny at home rather than allowing them to play outdoors [7]. Another factor that pertains to UAE in particular is the hot weather that persists throughout the year. Extreme heat and conditions that can cause heat strokes and dehydration in children prevent parents from sending them outside to get any physical activity. Irrespective to gender differences, it is essential to emphasize the importance of physical activity in young children and to increase awareness of parents

regarding its importance. Establishing adequate physical activity at a young age will help children maintain this habit throughout their life.

In the present study, more than half (54.9%) of the participants tend to have food of a mixed nature (carbohydrates, fats, and proteins). The majority of participants (87%) reported consuming food with a mixed preparation pattern (boiled, fried, and grilled). It has been noted that a minor percentage of the population consume either fats (1.6%) or carbohydrates (13.1%). Furthermore, more than one quarter (30.4%) of the participants reported consuming protein as their main food with 7.5% of this group being obese and overweight.

A study in the US found a correlation between BMI and dietary protein intake in children up to the age of 5 years [25]. High protein intake may result in a higher than normal BMI, this can be explained by the "Early Protein Hypothesis" which assumes that the high protein consumption will lead to high insulin-releasing amino acids in the plasma which will trigger insulin secretion and insulin like growth factor I (IGF-I) and leads to fat deposition and weight gain. It is worth noting that skipping breakfast is common among both adults and children, and this practice has a negative impact on BMI [14]. A study preformed in Al Ain, UAE reinforced that children who skipped breakfast are more likely to be obese [16]. According to the present results, about one quarter (21.1%) of the participants preferred eating snacks as their breakfast meal, which is mostly full of carbohydrates and with no beneficial nutritional value, and 1.8% of the participants skip breakfast. Collectively 13.6% of girls and 9.2% of boys of the total population have a habit of skipping a proper healthy breakfast meal. A systemic review carried out in Europe showed that individuals who eat breakfast are less likely to become obese or overweight [26]. In harmony with results of the present study, an earlier study in UAE reported that skipping breakfast is more common in girls (37%) as compared to boys (28%) in UAE [16].

Obesity has lately been associated with high consumption of fast food. Children often prefer fast food restaurants; and consumption of fast food has increased due to its convenience and affordability particularly in children of working parents. Regular consumption of fast food has shown bad influence on health as it contains a large number of calories with very low nutritional value, especially for children, who require nutrients for their growth and development. Despite the fact that numerous studies have showed that consumption of fast food causes weight gain, a relationship between the two factors is difficult to establish [4].

We pinpoint two methodological limitations that should be considered when interpreting the findings of the present study. First, we assessed dietary patterns by asking broad questions about food groups consumed and their quantity and sources of food. Hence, responses of participants may be subject to recall bias and personal interpretation than the use of a food diary that enables objective and detailed documentation of types and quantity of foods consumed within a specified period. Second, the UAE population is highly diverse, with the majority of people being expatriates from various Arabic-speaking, and South- and East-Asian countries, with Emirati citizens representing a minority. The marked differences in income, dietary practices, and cultural identities of the various ethnic groups in UAE can influence the development of obesity in children. In addition, Emirati children largely attend governmental schools while children of expatriates attend private schools. We did not specifically measure these factors in the present study and thus they require further investigation in large cross-sectional studies in the future.

## Conclusion

The percentage of overweight and obesity is high in both genders and across all ages of the studied population. However, we observed that the highest percentage of overweight is among

children at the age of 11years and this may be a consequence of their sedentary lifestyle, consumption of unhealthy food and family history. Further investigation regarding this topic is required, and increasing awareness of the public towards childhood obesity is to be achieved in the near future as an essential task to limit further progression of obesity in UAE and to protect against non-communicable diseases that may be associated with obesity.

## Supporting information

**S1 File.**
(PDF)

**S2 File.**
(PDF)

**S3 File.**
(PDF)

**S4 File.**
(PDF)

## Author Contributions

**Conceptualization:** Abduelmula R. Abduelkarem, Amar M. Hamrouni.

**Data curation:** Farah G. Bankessli, Sherin A. Kamal, Nahed M. Kulhasan.

**Formal analysis:** Abduelmula R. Abduelkarem.

**Investigation:** Abduelmula R. Abduelkarem.

**Methodology:** Abduelmula R. Abduelkarem, Farah G. Bankessli, Sherin A. Kamal, Nahed M. Kulhasan.

**Project administration:** Abduelmula R. Abduelkarem.

**Resources:** Farah G. Bankessli, Nahed M. Kulhasan.

**Supervision:** Abduelmula R. Abduelkarem.

**Validation:** Suleiman I. Sharif, Amar M. Hamrouni.

**Writing – original draft:** Abduelmula R. Abduelkarem, Farah G. Bankessli, Sherin A. Kamal, Nahed M. Kulhasan, Amar M. Hamrouni.

**Writing – review & editing:** Abduelmula R. Abduelkarem, Suleiman I. Sharif, Amar M. Hamrouni.

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
