## [Decision Letter · Decision Letter 0]

6 Apr 2020

PONE-D-19-26655

Obesity and its associated risk factors among school-aged children in Sharjah, UAE

PLOS ONE

Dear Dr Abduelkarem,

Thank you for submitting your manuscript to PLOS ONE. After careful consideration, we feel that it has merit but does not fully meet PLOS ONE’s publication criteria as it currently stands. Therefore, we invite you to submit a revised version of the manuscript that addresses the points raised during the review process.

Please make sure to address all the reviewers' comments.

We would appreciate receiving your revised manuscript by May 21 2020 11:59PM. To enhance the reproducibility of your results, we recommend that if applicable you deposit your laboratory protocols in protocols.io, where a protocol can be assigned its own identifier (DOI) such that it can be cited independently in the future. For instructions see: http://journals.plos.org/plosone/s/submission-guidelines#loc-laboratory-protocols

We look forward to receiving your revised manuscript.

Kind regards,

Robert Siegel

Academic Editor

PLOS ONE

2. Please include additional information regarding the survey or questionnaire used in the study and ensure that you have provided sufficient details that others could replicate the analyses. For instance, if you developed a questionnaire as part of this study and it is not under a copyright more restrictive than CC-BY, please include a copy, in both the original language and English, as Supporting Information. Moreover, please include more details on how the questionnaire was pre-tested, and whether it was validated.

Reviewers' comments:

Reviewer's Responses to Questions

**Comments to the Author**

1. Is the manuscript technically sound, and do the data support the conclusions?

Reviewer #1: Partly

Reviewer #2: Partly

2. Has the statistical analysis been performed appropriately and rigorously? 

Reviewer #1: N/A

Reviewer #2: I Don't Know

3. Have the authors made all data underlying the findings in their manuscript fully available?

Reviewer #1: Yes

Reviewer #2: Yes

4. Is the manuscript presented in an intelligible fashion and written in standard English?

Reviewer #1: Yes

Reviewer #2: Yes

5. Review Comments to the Author

Reviewer #1: Comments to the Author

This Study aimed to assess “obesity and its associated risk factors among school-aged children in sharjah”

1- In the introduction section the authors refer to the prevalence of childhood overweight and obesity in the US and worldwide, but they did not provide any description about the prevalence of childhood overweight and obesity in the UAE. Please add this data in the introduction.

2- Anthropometric measurements: Quality assurance for anthropocentric measurements should be described in details.

3- Is the questionnaire of dietary habit reliable or valid for this population study? Please provide more information about reliability and validity of this questionnaire.

4- The author did not provide any information about the physical activity in children. Please add this information in the methods section (questionnaire development section).

5- Based on method section you select population between native and non-native students, but in the results you did not refer to this. How many of the subjects were non-native? Are there any differences in the prevalence of childhood overweight and obesity between the native and non-native students?

6- What is the definition of physical inactivity that you reported in the result section?

7- In the statistical analysis the author used the chi-square test to ascertain the association between the dependent and independent variables, how you can use Chi-square test for ascertain an association? Please check the type of statistical tests with a statistician.

8- Table section: The author must mention that the type of statistical tests was used for assessing the difference of various socio-demographic, lifestyle, anthropometric, in across BMI percentile.

9- In the table you did not refer that what type of data was used, number or percent?

Reviewer #2: I commend the authors on taking on an important project. This type of research is needed in order to reverse the obesity epidemic that threatens all nations. I have multiple concerns about this manuscript. The methods section is missing pertinent details about how the sample was created (how many schools, every child in the age range within the school or certain grades, etc.; what are the demographics of the schools included – similar, different). Additionally, some of the questions in the survey are hard to follow (see line corrections below). The descriptive results in table 1 are not presented by BMI category, which makes it harder for the reader to understand the sample. There is insufficient text supporting the findings in table 4 & 5. Additionally, the discussion is too long. The authors should seek to identify the take home findings and what adds to the literature, rather than discussing every result. Also, a re-review of the grammar would be helpful, as there are multiple errors throughout. Lastly, some of the links provided within the references are outdated and need to be revisited and reinserted. Please see some additional line suggestions below.

Line suggestions

Abstract:

1. Line 36: Should read “…and determine” rather than determining

2. Line 38: insert “was” before the word based

3. Line 43: Recommend person-first language. Would recommend from “were obese or overweight” to “had obesity or overweight.”

Introduction:

4. Line 65-66: The consumption of high calories food with no or limited physical activity is the main contributor to childhood obesity. This statement needs a citation.

5. Line 73: Run on sentence, needs appropriate punctuation.

6. Line 80: Two strategies can be followed including; the primary prevention of overweight

or obesity itself and second is the prevention of weight regain after weight loss, plus prevention of weight gain in individuals unable to lose weight [5]. This statement is confusing. Initially talks about two strategies, but introduces a third.

7. Line 100: Add “the” before the word questionnaire.

Methods:

8. The survey question asking the Nature of food consumed to parents is very confusing. Why were fruit and vegetables not included as an answer choice for the question: predominant food in child’s diet? What does mixed mean – 2 categories, 3 categories? Would be better to have had families give percentages. Was fruit included in carbohydrate? Was the any ascertainment of baseline knowledge level – were families clear on what food groups fell into each of these categories?

9. I am concerned about selection bias. The parents/child that returned the survey could be inherently different that the ones that didn’t. Did you have any data to say that your sample population is representative? Do you have any data on the parents/children that chose not to participate or the data on the schools as a whole?

Study Population

10. How many schools were participants recruited from? Were the schools the same in terms of demographics of students?

11. Were the surveys sent to every child in a specific grade. Need to expand on how the sample was put together in more detail.

Results:

12. Line 153: Person first language recommended. (E.g. Had an obese BMI, rather than were obese.)

13. Descriptive characteristics should be presented by BMI category.

14. It is unclear how disease status and medication use were related to the aims of the study.

6. PLOS authors have the option to publish the peer review history of their article (what does this mean?). If published, this will include your full peer review and any attached files.

Reviewer #1: No

Reviewer #2: No

---

## [Author Response · Author response to Decision Letter 0]

7 May 2020

All the invaluable comments of the respected reviewers and the journal additional comments were considered and our responses are attached as “our responses to reviewer’s comments” and “our responses to additional journal comments” two separate file.

As per your request, a copy of the survey, in both the original language (Arabic) and English, will be attached for you as Supporting Information. 

To answer your request regarding “copyedit your manuscript for language usage, spelling, and grammar”, the manuscript was thoroughly revised by one of us namely Prof. Suleiman I. Sharif, College of Pharmacy, University of Sharjah, Sharjah, UAE. 

A copy of our manuscript showing our changes by either highlighting them or using track changes will be attached for you as revised version and clean copy of the edited manuscript and it will be uploaded as a “supporting information” file.

We would like to clearly indicate that there are no ethical or legal restrictions on sharing the data. It must be noted that in our study the identity of the participants was disguised.

---

## [Decision Letter · Decision Letter 1]

22 May 2020

Obesity and its associated risk factors among school-aged children in Sharjah, UAE

PONE-D-19-26655R1

Dear Dr. Abduelkarem,

We are pleased to inform you that your manuscript has been judged scientifically suitable for publication and will be formally accepted for publication once it complies with all outstanding technical requirements.

With kind regards,

Robert Siegel

Academic Editor

PLOS ONE

Additional Editor Comments (optional):

The authors have addressed all the reviewer concerns.

Reviewers' comments:

Reviewer's Responses to Questions

**Comments to the Author**

1. If the authors have adequately addressed your comments raised in a previous round of review and you feel that this manuscript is now acceptable for publication, you may indicate that here to bypass the “Comments to the Author” section, enter your conflict of interest statement in the “Confidential to Editor” section, and submit your "Accept" recommendation.

Reviewer #1: All comments have been addressed

Reviewer #2: All comments have been addressed

2. Is the manuscript technically sound, and do the data support the conclusions?

Reviewer #1: Yes

Reviewer #2: (No Response)

3. Has the statistical analysis been performed appropriately and rigorously? 

Reviewer #1: Yes

Reviewer #2: (No Response)

4. Have the authors made all data underlying the findings in their manuscript fully available?

Reviewer #1: Yes

Reviewer #2: (No Response)

5. Is the manuscript presented in an intelligible fashion and written in standard English?

Reviewer #1: Yes

Reviewer #2: (No Response)

6. Review Comments to the Author

Reviewer #1: this manuscript accepted without revision

Reviewer #2: (No Response)

7. PLOS authors have the option to publish the peer review history of their article (what does this mean?). If published, this will include your full peer review and any attached files.

Reviewer #1: No

Reviewer #2: No

---

## [Editor Report · Acceptance letter]

28 May 2020

PONE-D-19-26655R1 

Obesity and its associated risk factors among school-aged children in Sharjah, UAE 

Dear Dr. Abduelkarem:

I am pleased to inform you that your manuscript has been deemed suitable for publication in PLOS ONE. Congratulations! Your manuscript is now with our production department. 

With kind regards,

on behalf of

Dr. Robert Siegel 

Academic Editor

PLOS ONE